## [Peer Review File · Nature Immunology]

CD38 endows local antigen-specific Treg cells with stress resilience for control of compartmentalized CNS inflammation

Corresponding Author: Dr Thomas Korn

Version 0:

Reviewer comments:

Reviewer #1

(Remarks to the Author)

In their manuscript 'CD38 endows local antigen-specific Foxp3+ Treg cells with stress resilience necessary to exert compartmentalized control over chronic CNS inflammation', Chen and colleagues investigate the role of Treg cells after CNS inflammation. They characterized Treg cells during EAE peak and recovery phase and, using transfer experiments, could show that CD38 expression on Treg cells is required for the regulatory capacity of Treg cells during EAE.

Mechanistically, they suggest that CD38 consumes NAD which prevents the ADP-ribosylation of cell surface molecules (including the IL2 receptor) and allows for IL-2 signaling which is vital for Treg cells.

Overall, this study provides a very important contribution towards better understanding the role of tissue-resident Treg cell for local immune homeostasis after an inflammatory event. However, the following concerns should be addressed before publication:

Major concerns:

- 1) Figure 2 includes an interesting data set showing that the local depletion of CNS-Tregs results in the re-establishment of EAE symptoms. Here, the authors only look until 3 days after Treg depletion. However, it would be interesting to also investigate later time points. Can immune homeostasis and EAE recovery be re-established when the Treg cells return?
- 2) In Figure 3, the authors state that their 'stress tolerant' CNS Treg cells resemble tissue-Treg cells. They show that this tissue Treg signature is present both in peak and in recovery phase CNS Treg cells compared to their splenic counterparts and define a recovery phase CNS Treg signature. However, to truly state that any changes observed are caused by recovery from EAE and don't simply represent a return to homeostasis, a control consisting of naive CNS Treg cells is required. Are the authors certain that the parts of the signature they use to define their Treg population as 'stress tolerant' is not a part of the tissue Treg signature? Perhaps it would also be interesting to compare recovery phase Treg cells to bona fide tissue Treg cells from a different organ.
- 3) Where are the stress-resistant CNS Treg cells located in the brain? It would be interesting to see histological images showing their location.
- 4) In Figure 7, the authors generated mixed bone marrow chimeras with CD38 KO and WT Tregs and Foxp3-DTR Treg cells. After disease peak, the authors depleted the WT Foxp3-DTR cells, leaving only either WT or CD38 KO Treg cells behind and observed that animals with CD38 deficient Treg cells deteriorate more quickly. However, the authors do not depict disease score but rather the Δ EAE score. It would be interesting to see whether/how the disease progression differs between the different bone marrow chimeras (i.e., is the initial disease score before depletion of the WT Treg cells different between the two groups?).

Minor concerns:

- 5) Figure 7b: please include a quantification of the depletion efficiency.
- 6) Structure of the manuscript: In Figure 6, the authors elucidate the mechanism behind their observation that CD38 is crucial for the regulatory capacity of Treg cells in the CNS. Figure 7, however, only shows (again) that a loss of CD38 leads to reduced protection against EAE. Perhaps changing the order of these two figures and ending on unraveling the mechanism behind their observations would make for a more compelling read.
- 7) In the abstract, the authors introduce the term 'inflammation tolerant' to describe their Treg cells which they never mention again. Instead, they pivot to 'stress tolerant' in their main text. Perhaps this could be harmonized.
- 8) Continuously misspelled score as sore in the figure panels depicting EAE scores.

Reviewer #2

(Remarks to the Author)

This manuscript by Chen et al investigates the mechanisms by which Tregs survive in inflammatory tissues, in particular the inflamed brain. This environment is rich in NAD⁺ (which can ADP-ribosylate CD25, decreasing signalling capacity), thus potentially requiring counter-acting factors such as CD38 expression. The experiments presented here demonstrate the role of CNS-resident Tregs in preventing relapses during the recovery phase, with the upregulation of CD38 demonstrated to be a key molecular event to enable the survival of CNS-resident Tregs during this inflamed state. There are limitations to the approaches used, however these are all clearly articulated. The work is an important contribution to the fields of tissue immunity, neuroinflammation and regulatory T cells.

Minor issues:

The authors perform an elegant experiment of photoactivating T cells from the draining lymph node of the site of MOG injection, and show that CNS-Tregs during the resolution phase are largely not coming from this lymph node (Figure 1e). The authors interpret this to mean that the CNS-Tregs at this stage are largely disconnected from systemic immunity, however it would be more precise to say that they are disconnected from the original T cell priming point, as this experiment does not exclude Tregs from the rest of the body as being a major source of incoming cells. The very low proliferation rate of CNS Tregs at this stage would be consistent with an external source incoming.

The text mentions that CD38 is selectively upregulated in recovery Tregs (Fig 4a). It would be more accurate to say that CD38 is upregulated in peak disease Tregs, and further upregulated in recovery Tregs.

The interpretation of the data in Fig 5e-f is rather strong, considering the subtle effects and high variation. The magnitude of the effect in Tconv and Treg is similar (albeit in opposite directions), and the effect of CD38 loss on the MOG-reactive Tregs is limited. This should be considered in the textual description of the effect, perhaps inserting the word "partially" when describing the Tregs as being CD38 dependent. Considering the strength of the clinical impact of CD38 depletion, do the authors exclude a functional role of CD38 beyond maintaining the CNS Treg compartment?

If CD38 works via preserving CD25 signalling, would enhancing the local IL2 levels in the CNS rescue the loss of Tregs observed in a CD38 KO?

Reviewer #3

(Remarks to the Author)

The manuscript by Chen et al., investigates the role of Foxp3⁺ regulatory T cells (Treg) in the central nervous system (CNS) following the initial wave of inflammation in experimental autoimmune encephalomyelitis (EAE). Using an impressive array of molecular approaches and an important method for preferentially depleting peripheral versus CNS-resident Tregs, the authors demonstrate that these Tregs have a distinct phenotype, persist in the CNS, are essential to prevent the re-initiation CNS inflammation by residual effector T cells, and that part of their maintenance and suppressive program in the CNS is related to CD38 regulation of their surface IL-2R expression.

Tregs have been previously known recruited to the CNS during the active phase of EAE and are thought to play a role in the initial resolution of active disease. However, the role of CNS Tregs in the post-acute phase of inflammation has not been previously known. How chronic inflammation persists and is regulated in autoimmune disease tissues is increasingly becoming of paramount importance in the context of Multiple Sclerosis (MS) and other chronic inflammatory disorders. Addressing the comments below should improve the manuscript.

Major Concerns

1. The clinical score curves used after DTx treatment (such as in Figure 2d) are unusual for the field. It would be helpful if the authors could show one example full EAE score curve from Day 0 to last time point, to visualize how the DTx i.c. treatment impacts clinical scores. Additionally, histology images from DTx i.c. treated vs control mice would help validate that there is increased inflammation after local Treg depletion. This could be included in supplemental data.

Minor Concerns

1. The authors use the term "MOG-activated" Tregs in the Fig. 4b experimental setup. They might wish to state that the experimental setup in Fig 4b "enriches" for MOG reactive Tregs, as the vast majority of transferred Tregs from the spleen in this setup will not be MOG-reactive.

2. On page 18, the authors argue that the effect of CD38 on Tregs was more pronounced in the antigen-specific cells based on their tetramer data. This is likely the case. In addition, tetramer staining intensity correlates with TCR affinity for a pMHC complex, and it has been shown that many non-MOG tetramer binding T cells in the CNS of EAE are have TCRs with low affinity for MOG(35-55)-I-Ab. Thus, the authors may wish to mention that CD38 could also be more important on Tregs with a TCR that is high-affinity for MOG(35-55)-I-Ab.

3. The authors note and provide compelling data for negligible exchange of "inflammation-tolerant CNS Tregs" with the periphery. To further support these data, they may wish to cite Wakim et al., PNAS (2010) which similarly showed that CD8⁺ T resident memory cells in the CNS adapt to the CNS environment, remain for long periods even without cognate antigen, and rapidly die when removed from the CNS milieu.

Decision Letter:

16th Sep 2025

Dear Dr Korn,

Your Article, "CD38 endows local antigen-specific Foxp3+ Treg cells with stress resilience necessary to exert compartmentalized control over chronic CNS inflammation" has now been seen by 3 referees.

You will see from their comments copied below that while they find your work of considerable interest, they have raised a few concerns that must be addressed. In light of these comments, we cannot accept the manuscript for publication, but would be very interested in considering a revised version that addresses these concerns.

If you choose to revise your manuscript, please highlight all changes in the manuscript text file in Microsoft Word format.

* If you have not done so already please begin to revise your manuscript so that it conforms to our Article format instructions at <http://www.nature.com/ni/authors/index.html>. Refer also to any guidelines provided in this letter.

The Reporting Summary can be found here:

Extended Data figures and tables are online-only (appearing in the online PDF and full-text HTML version of the paper), peer-reviewed display items that provide essential background to the Article but are not included in the printed version of the paper due to space constraints or being of interest only to a few specialists. A maximum of ten Extended Data display items (figures and tables) is typically permitted. When re-submitting your manuscript, please ensure that any supplementary figures and tables that are more critical to the manuscript's conclusions are converted to Extended data to increase these data's visibility.

Link Redacted

If you wish to submit a suitably revised manuscript we would hope to receive it within 6 months. If you cannot send it within this time, please let us know. We will be happy to consider your revision so long as nothing similar has been accepted for publication at Nature Immunology or published elsewhere.

Nature Immunology is committed to improving transparency in authorship. As part of our efforts in this direction, we are now requesting that all authors identified as 'corresponding author' on published papers create and link their Open Researcher and Contributor Identifier (ORCID) with their account on the Manuscript Tracking System (MTS), prior to acceptance. ORCID

helps the scientific community achieve unambiguous attribution of all scholarly contributions. You can create and link your ORCID from the home page of the MTS by clicking on 'Modify my Springer Nature account'. For more information please visit www.springernature.com/orcid.

Thank you for the opportunity to review your work.

Sincerely,

Nick Bernard, PhD
Senior Editor
Nature Immunology

Reviewers' Comments:

Reviewer #1 (Remarks to the Author):

In their manuscript 'CD38 endows local antigen-specific Foxp3+ Treg cells with stress resilience necessary to exert compartmentalized control over chronic CNS inflammation', Chen and colleagues investigate the role of Treg cells after CNS inflammation. They characterized Treg cells during EAE peak and recovery phase and, using transfer experiments, could show that CD38 expression on Treg cells is required for the regulatory capacity of Treg cells during EAE. Mechanistically, they suggest that CD38 consumes NAD which prevents the ADP-ribosylation of cell surface molecules (including the IL2 receptor) and allows for IL-2 signaling which is vital for Treg cells. Overall, this study provides a very important contribution towards better understanding the role of tissue-resident Treg cell for local immune homeostasis after an inflammatory event. However, the following concerns should be addressed before publication:

Major concerns:

- 1) Figure 2 includes an interesting data set showing that the local depletion of CNS-Tregs results in the re-establishment of EAE symptoms. Here, the authors only look until 3 days after Treg depletion. However, it would be interesting to also investigate later time points. Can immune homeostasis and EAE recovery be re-established when the Treg cells return?
- 2) In Figure 3, the authors state that their 'stress tolerant' CNS Treg cells resemble tissue-Treg cells. They show that this tissue Treg signature is present both in peak and in recovery phase CNS Treg cells compared to their splenic counterparts and define a recovery phase CNS Treg signature. However, to truly state that any changes observed are caused by recovery from EAE and don't simply represent a return to homeostasis, a control consisting of naive CNS Treg cells is required. Are the authors certain that the parts of the signature they use to define their Treg population as 'stress tolerant' is not a part of the tissue Treg signature? Perhaps it would also be interesting to compare recovery phase Treg cells to bona fide tissue Treg cells from a different organ.
- 3) Where are the stress-resistant CNS Treg cells located in the brain? It would be interesting to see histological images showing their location.
- 4) In Figure 7, the authors generated mixed bone marrow chimeras with CD38 KO and WT Tregs and Foxp3-DTR Treg cells. After disease peak, the authors depleted the WT Foxp3-DTR cells, leaving only either WT or CD38 KO Treg cells behind and observed that animals with CD38 deficient Treg cells deteriorate more quickly. However, the authors do not depict disease score but rather the Δ EAE score. It would be interesting to see whether/how the disease progression differs between the different bone marrow chimeras (i.e., is the initial disease score before depletion of the WT Treg cells different between the two groups?).

Minor concerns:

- 5) Figure 7b: please include a quantification of the depletion efficiency.
- 6) Structure of the manuscript: In Figure 6, the authors elucidate the mechanism behind their observation that CD38 is crucial for the regulatory capacity of Treg cells in the CNS. Figure 7, however, only shows (again) that a loss of CD38 leads to reduced protection against EAE. Perhaps changing the order of these two figures and ending on unraveling the mechanism behind their observations would make for a more compelling read.
- 7) In the abstract, the authors introduce the term 'inflammation tolerant' to describe their Treg cells which they never mention again. Instead, they pivot to 'stress tolerant' in their main text. Perhaps this could be harmonized.
- 8) Continuously misspelled score as sore in the figure panels depicting EAE scores.

Reviewer #2 (Remarks to the Author):

This manuscript by Chen et al investigates the mechanisms by which Tregs survive in inflammatory tissues, in particular the inflamed brain. This environment is rich in NAD⁺ (which can ADP-ribosylate CD25, decreasing signalling capacity), thus potentially requiring counter-acting factors such as CD38 expression. The experiments presented here demonstrate the role of CNS-resident Tregs in preventing relapses during the recovery phase, with the upregulation of CD38 demonstrated to be a key molecular event to enable the survival of CNS-resident Tregs during this inflamed state. There are limitations to the

approaches used, however these are all clearly articulated. The work is an important contribution to the fields of tissue immunity, neuroinflammation and regulatory T cells.

Minor issues:

The authors perform an elegant experiment of photoactivating T cells from the draining lymph node of the site of MOG injection, and show that CNS-Tregs during the resolution phase are largely not coming from this lymph node (Figure 1e). The authors interpret this to mean that the CNS-Tregs at this stage are largely disconnected from systemic immunity, however it would be more precise to say that they are disconnected from the original T cell priming point, as this experiment does not exclude Tregs from the rest of the body as being a major source of incoming cells. The very low proliferation rate of CNS Tregs at this stage would be consistent with an external source incoming.

The text mentions that CD38 is selectively upregulated in recovery Tregs (Fig 4a). It would be more accurate to say that CD38 is upregulated in peak disease Tregs, and further upregulated in recovery Tregs.

The interpretation of the data in Fig 5e-f is rather strong, considering the subtle effects and high variation. The magnitude of the effect in Tconv and Treg is similar (albeit in opposite directions), and the effect of CD38 loss on the MOG-reactive Tregs is limited. This should be considered in the textual description of the effect, perhaps inserting the word "partially" when describing the Tregs as being CD38 dependent. Considering the strength of the clinical impact of CD38 depletion, do the authors exclude a functional role of CD38 beyond maintaining the CNS Treg compartment?

If CD38 works via preserving CD25 signalling, would enhancing the local IL2 levels in the CNS rescue the loss of Tregs observed in a CD38 KO?

Reviewer #3 (Remarks to the Author):

The manuscript by Chen et al., investigates the role of Foxp3+ regulatory T cells (Treg) in the central nervous system (CNS) following the initial wave of inflammation in experimental autoimmune encephalomyelitis (EAE). Using an impressive array of molecular approaches and an important method for preferentially depleting peripheral versus CNS-resident Tregs, the authors demonstrate that these Tregs have a distinct phenotype, persist in the CNS, are essential to prevent the re-initiation CNS inflammation by residual effector T cells, and that part of their maintenance and suppressive program in the CNS is related to CD38 regulation of their surface IL-2R expression.

Tregs have been previously known recruited to the CNS during the active phase of EAE and are thought to play a role in the initial resolution of active disease. However, the role of CNS Tregs in the post-acute phase of inflammation has not been previously known. How chronic inflammation persists and is regulated in autoimmune disease tissues is increasingly becoming of paramount importance in the context of Multiple Sclerosis (MS) and other chronic inflammatory disorders. Addressing the comments below should improve the manuscript.

Major Concerns

1. The clinical score curves used after DTx treatment (such as in Figure 2d) are unusual for the field. It would be helpful if the authors could show one example full EAE score curve from Day 0 to last time point, to visualize how the DTx i.c. treatment impacts clinical scores. Additionally, histology images from DTx i.c. treated vs control mice would help validate that there is increased inflammation after local Treg depletion. This could be included in supplemental data.

Minor Concerns

1. The authors use the term "MOG-activated" Tregs in the Fig. 4b experimental setup. They might wish to state that the experimental setup in Fig 4b "enriches" for MOG reactive Tregs, as the vast majority of transferred Tregs from the spleen in this setup will not be MOG-reactive.

2. On page 18, the authors argue that the effect of CD38 on Tregs was more pronounced in the antigen-specific cells based on their tetramer data. This is likely the case. In addition, tetramer staining intensity correlates with TCR affinity for a pMHC complex, and it has been shown that many non-MOG tetramer binding T cells in the CNS of EAE are have TCRs with low affinity for MOG(35-55)-I-Ab. Thus, the authors may wish to mention that CD38 could also be more important on Tregs with a TCR that is high-affinity for MOG(35-55)-I-Ab.

3. The authors note and provide compelling data for negligible exchange of "inflammation-tolerant CNS Tregs" with the periphery. To further support these data, they may wish to cite Wakim et al., PNAS (2010) which similarly showed that CD8+ T resident memory cells in the CNS adapt to the CNS environment, remain for long periods even without cognate antigen, and rapidly die when removed from the CNS milieu.

Version 1:

Reviewer comments:

Reviewer #1

(Remarks to the Author)

All of my questions have been successfully addressed by the authors.

This study offers a significant contribution to advancing the understanding of the role of tissue Treg cells in maintaining local immune homeostasis following an inflammatory event.

Reviewer #3

(Remarks to the Author)

The revised manuscript by Chen et al., investigates the role of Foxp3+ regulatory T cells (Treg) in the central nervous system (CNS) following the initial wave of inflammation in experimental autoimmune encephalomyelitis (EAE). The authors have now definitively demonstrated that these Tregs have a distinct phenotype, persist in the CNS, prevent the re-initiation of CNS inflammation by residual effector T cells, and that CD38 regulation of IL-2R expression is important for their function. Tregs have been previously known to be recruited to the CNS during the active phase of EAE and are thought to play a role in the initial resolution of active disease. However, the role of CNS Tregs in the post-acute phase of inflammation has not been previously known.

The authors have made an impressive number of revisions and have addressed all my prior concerns. Notably, the histology in Figure 3 and the different visualizations of clinical scores following i.c. DTX as shown in Supplementary Figure 3 highlight the striking phenotype the authors have observed. These data clearly show that Tregs contribute both the re-establishment and maintenance of immune homeostasis in the CNS after autoimmune attack.

Major Concerns

1. none

Minor Concerns

1. none

Decision Letter:

Our ref: NI-A41145A

18th Nov 2025

Dear Dr. Korn,

Thank you for submitting your revised manuscript "CD38 endows local antigen-specific Foxp3+ Treg cells with stress resilience necessary to exert compartmentalized control over chronic CNS inflammation" (NI-A41145A). It has now been seen by the original referees and their comments are below. The reviewers find that the paper has improved in revision, and therefore we'll be happy in principle to publish it in Nature Immunology, pending minor revisions to comply with our editorial and formatting guidelines.

We will now perform detailed checks on your paper and will send you a checklist detailing our editorial and formatting requirements in about a week. Please do not upload the final materials and make any revisions until you receive this additional information from us.

If you had not uploaded a Word file for the current version of the manuscript, we will need one before beginning the editing process; please email that to immunology@us.nature.com at your earliest convenience.

Thank you again for your interest in Nature Immunology. Please do not hesitate to contact me if you have any questions.

Sincerely,

Nick Bernard, PhD
Senior Editor
Nature Immunology

Reviewer #1 (Remarks to the Author):

All of my questions have been successfully addressed by the authors.

This study offers a significant contribution to advancing the understanding of the role of tissue Treg cells in maintaining local immune homeostasis following an inflammatory event.

Reviewer #3 (Remarks to the Author):

The revised manuscript by Chen et al., investigates the role of Foxp3+ regulatory T cells (Treg) in the central nervous system (CNS) following the initial wave of inflammation in experimental autoimmune encephalomyelitis (EAE). The authors have

now definitively demonstrated that these Tregs have a distinct phenotype, persist in the CNS, prevent the re-initiation CNS inflammation by residual effector T cells, and that CD38 regulation of IL-2R expression is important for their function. Tregs have been previously known to be recruited to the CNS during the active phase of EAE and are thought to play a role in the initial resolution of active disease. However, the role of CNS Tregs in the post-acute phase of inflammation has not been previously known.

The authors have made an impressive number of revisions and have addressed all my prior concerns. Notably, the histology in Figure 3 and the different visualizations of clinical scores following i.c. DTX as shown in Supplementary Figure 3 highlight the striking phenotype the authors have observed. These data clearly show that Treg contribute both the re-establishment and maintenance of immune homeostasis in the CNS after autoimmune attack.

Major Concerns

1. none

Minor Concerns

1. none

Version 2:

Decision Letter:

In reply please quote: NI-A41145B

Dear Dr. Korn,

I am delighted to accept your manuscript entitled "CD38 endows local antigen-specific Treg cells with stress resilience for control of compartmentalized CNS inflammation" for publication in an upcoming issue of Nature Immunology. Happy NYE.

Over the next few weeks, your paper will be copyedited to ensure that it conforms to Nature Immunology style. Once your paper is typeset, you will receive an email with a link to choose the appropriate publishing options for your paper and our Author Services team will be in touch regarding any additional information that may be required.

Authors may need to take specific actions to achieve compliance with funder and institutional open access mandates. If your research is supported by a funder that requires immediate open access (e.g. according to <https://www.springernature.com/gp/open-science/plan-s-compliance> Plan S principles or the <https://www.springernature.com/gp/open-science/us-federal-agency-compliance> NIH public access policy) then you should select the gold OA route, and we will direct you to the compliant route where possible. Because authors warrant under our subscription licensing terms that they haven't committed to licensing any version of their article under a licence inconsistent with the terms of our agreement – including the applicable embargo period – publication under the subscription model isn't suitable for authors whose funders require no embargo.

Your paper will be published online soon after we receive your corrections and will appear in print in the next available issue.

You may wish to make your media relations office aware of your accepted publication, in case they consider it appropriate to organize some internal or external publicity. Once your paper has been scheduled you will receive an email confirming the

publication details. This is normally 3-4 working days in advance of publication. If you need additional notice of the date and time of publication, please let the production team know when you receive the proof of your article to ensure there is sufficient time to coordinate. Further information on our embargo policies can be found here:
<https://www.nature.com/authors/policies/embargo.html>

Also, if you have any spectacular or outstanding figures or graphics associated with your manuscript - though not necessarily included with your submission - we'd be delighted to consider them as candidates for our cover. Simply send an electronic version (accompanied by a hard copy) to us with a possible cover caption enclosed.

If you have not already done so, we strongly recommend that you upload the step-by-step protocols used in this manuscript to protocols.io. protocols.io is an open online resource that allows researchers to share their detailed experimental know-how. All uploaded protocols are made freely available and are assigned DOIs for ease of citation. Protocols can be linked to any publications in which they are used and will be linked to from your article. You can also establish a dedicated workspace to collect all your lab Protocols. By uploading your Protocols to protocols.io, you are enabling researchers to more readily reproduce or adapt the methodology you use, as well as increasing the visibility of your protocols and papers. Upload your Protocols at <https://protocols.io>. Further information can be found at <https://www.protocols.io/help/publish-articles>.

Please note that we encourage the authors to self-archive their manuscript (the accepted version before copy editing) in their institutional repository, and in their funders' archives, six months after publication. Nature Portfolio recognizes the efforts of funding bodies to increase access of the research they fund, and strongly encourages authors to participate in such efforts. For information about our editorial policy, including license agreement and author copyright, please visit www.nature.com/ni/about/ed_policies/index.html

Sincerely,

Nick Bernard, PhD
Senior Editor
Nature Immunology

Click here if you would like to recommend Nature Immunology to your librarian
<http://www.nature.com/subscriptions/recommend.html#forms>

** Visit the Springer Nature Editorial and Publishing website at http://editorial-jobs.springernature.com?utm_source=ejP_NImm_email&utm_medium=ejP_NImm_email&utm_campaign=ejp_NImm for more information about our career opportunities. If you have any questions please click [here](mailto:editorial.publishing.jobs@springernature.com).

Dear Dr. Bernard,

Thank you for allowing us to revise our manuscript. We would like to thank the reviewers for their constructive comments. According to their recommendations, we revised our manuscript and added additional data. Please find attached a detailed point-by-point response to the reviewers' comments and critiques. All changes in the revised manuscript are highlighted in yellow.

Reviewer #1:

1) Figure 2 includes an interesting data set showing that the local depletion of CNS-Tregs results in the re-establishment of EAE symptoms. Here, the authors only look until 3 days after Treg depletion. However, it would be interesting to also investigate later time points. Can immune homeostasis and EAE recovery be re-established when the Treg cells return?

This is an important question. We repeated the local Treg cell depletion experiment, including both control groups, i.e., i.p. injection of DTx and i.c. injection of DTx into EAE mice with non-depletable Treg cells, and followed the animals for up to 10 days after local depletion of CNS Treg cells. We found that the relapses induced by the depletion of CNS Treg cells were hardly controlled by the animals. In fact, the majority of DTx-treated mice succumbed to their relapses (Supplementary Fig. 3b), suggesting that the CNS Treg cell niche was not replenished sufficiently with antigen-specific Treg cells after local DTx-mediated depletion. We show these additional clinical data in the new Supplementary Fig. 3 of the revised manuscript (see also p. 8 of the revised manuscript).

2) In Figure 3, the authors state that their 'stress tolerant' CNS Treg cells resemble tissueTreg cells. They show that this tissue Treg signature is present both in peak and in recovery phase CNS Treg cells compared to their splenic counterparts and define a recovery phase CNS Treg signature. However, to truly state that any changes observed are caused by recovery from EAE and don't simply represent a return to homeostasis, a control consisting of naive CNS Treg cells is required. Are the authors certain that the parts of the signature they use to define their Treg population as 'stress tolerant' is not a part of the tissue Treg signature? Perhaps it would also be interesting to compare recovery phase Treg cells to bona fide tissue Treg cells from a different organ.

These are relevant comments. To address these points, we have now included a bulk RNA-seq dataset of Treg cells isolated from naive (unmanipulated) mice, as well as bulk RNA-seq data from Treg cells isolated from the visceral adipose tissue (VAT), in the revised version of the manuscript. Since the number of Foxp3+ Treg cells that can be isolated from a naive CNS is in the range of 50 cells per mouse, the sequencing depth of naive CNS Treg cells is not as deep as in the other CNS Treg cell data sets. However, the comparison of these data sets allows for a more comprehensive picture of the signature of recovery CNS Treg cells ("stress-tolerant" Treg cells). There is a partial overlap with a "tissue-type" Treg cell signature that has been defined in VAT Treg cells. Also, recovery CNS Treg cells are more similar to naive CNS Treg cells than peak CNS Treg cells. However, other features of stress-tolerant CNS Treg cells, such as their greater addiction to IL-2, are distinct from those of naive CNS Treg cells. To further illustrate the partial overlap of the recovery CNS Treg cell signature with a CNS tissue signature, we measured the expression

of a set of Treg cell-associated molecules using flow cytometry and analyzed them by FlowSOM analysis. These novel data are included in the new Fig. 5 and Supplementary Fig. 5 of the revised manuscript.

3) Where are the stress-resistant CNS Treg cells located in the brain? It would be interesting to see histological images showing their location.

We performed extensive histopathologic analyses of recovery CNS ("stress-tolerant") Treg cells. In addition to representative example microphotographs, we included a characterization of Treg cell hotspots in the vicinity of the third ventricle, where recovery Treg cells are consistently located, in addition to the meningeal compartment. These data are displayed in the new Supplementary Fig. 1. See also p. 6 of the revised manuscript.

4) In Figure 7, the authors generated mixed bone marrow chimeras with CD38 KO and WT Tregs and Foxp3-DTR Treg cells. After disease peak, the authors depleted the WT Foxp3DTR cells, leaving only either WT or CD38 KO Treg cells behind and observed that animals with CD38 deficient Treg cells deteriorate more quickly. However, the authors do not depict disease score but rather the Δ EAE score. It would be interesting to see whether/how the disease progression differs between the different bone marrow chimeras (i.e., is the initial disease score before depletion of the WT Treg cells different between the two groups?).

Wild-type plus "DTR" mixed bone marrow mice behave differently in terms of their overall EAE scores as compared with CD38 KO plus "DTR" mixed bone marrow chimeras. In order to illustrate this difference, we now show EAE curves of global *Cd38*^{-/-} mice and *Cd38*^{+/-} mice (Supplementary Fig. 6a). We also provide the complete EAE curves of the 'wild-type plus DTR' MBMC and '*Cd38*^{-/-} plus DTR' MBMC, in which the Tconv compartment is a 1:1 mix of wild-type and *Cd38*^{-/-} conventional T cells. These data are shown in the Supplementary Fig. 6b of the revised manuscript. The explanation of the experimental design is given in full detail on pp. 13/14 of the revised manuscript.

Minor concerns:

5) Figure 7b: Please include a quantification of the depletion efficiency.

Since we read out the MBMC experiment on day 3 after i.c. DTx injection, we cannot provide the actual depletion efficiency in the CNS on day 1 after i.c. DTx injection. However, in the new Figs. 6h-i, we show that on day 3 after local depletion of wild-type Treg cells by i.c. DTx injection, the CNS Treg cell compartment consisted mostly of CD38-deficient Treg cells in the *Cd38*^{-/-} MBMC test group as compared to the *Cd38*^{+/-} MBMC control group. These data demonstrate that the prior depletion of DTR+ wild-type Treg cells was efficient.

6) Structure of the manuscript: In Figure 6, the authors elucidate the mechanism behind their observation that CD38 is crucial for the regulatory capacity of Treg cells in the CNS. Figure 7, however, only shows (again) that a loss of CD38 leads to reduced protection against EAE. Perhaps changing the order of these two figures and ending on unraveling the mechanism behind their observations would make for a more compelling read.

We agree with this suggestion and restructured the revised version of the manuscript accordingly. The data from the previous Fig. 7 are now integrated as subpanels into Fig. 6 to illustrate that CD38 is particularly required to maintain immune tolerance after the peak of disease, when local sources of IL-2 are limited. The mechanistic experiments are depicted in Figures 7 and 8 of the revised manuscript.

7) In the abstract, the authors introduce the term 'inflammation tolerant' to describe their Treg cells which they never mention again. Instead, they pivot to 'stress-tolerant' in their main text. Perhaps this could be harmonized.

Thank you for this suggestion. We have harmonized the terminology to "stress-tolerant" Treg cells in the revised version of the manuscript.

8) Continuously misspelled score as sore in the figure panels depicting EAE scores.

We apologize for these misspellings. This has been fixed in the revised manuscript.

Reviewer #2:

Minor issues:

The authors perform an elegant experiment of photoactivating T cells from the draining lymph node of the site of MOG injection, and show that CNS-Tregs during the resolution phase are largely not coming from this lymph node (Figure 1e). The authors interpret this to mean that the CNS-Tregs at this stage are largely disconnected from systemic immunity, however it would be more precise to say that they are disconnected from the original T cell priming point, as this experiment does not exclude Tregs from the rest of the body as being a major source of incoming cells. The very low proliferation rate of CNS Tregs at this stage would be consistent with an external source incoming.

We thank this reviewer for his/her favorable evaluation of our study. It is correct to say the experimental design of this experiment allows for the conclusion that CNS Treg cells are disconnected from their initial priming site in the peripheral immune system. Accordingly, we rephrased our statement in the text. However, in other contexts, we have used a photoconversion system (Hiltensperger et al., 2021) to also label Treg cells in mesenteric lymph nodes. Very similar to Treg cells located in inguinal lymph nodes, mesenterically localized Treg cells do not traffic to the CNS in measurable amounts in late stages of EAE either.

The text mentions that CD38 is selectively upregulated in recovery Tregs (Fig 4a). It would be more accurate to say that CD38 is upregulated in peak disease Tregs, and further upregulated in recovery Tregs.

Correct, we have rephrased this in the revised version of the manuscript (see p. 12 of the revised manuscript).

The interpretation of the data in Fig 5e-f is rather strong, considering the subtle effects and high variation. The magnitude of the effect in Tconv and Treg is similar (albeit in opposite directions), and the effect of CD38 loss on the MOG-reactive Tregs is limited. This should be considered in the textual description of the effect, perhaps inserting the word "partially" when describing the Tregs as being CD38 dependent. Considering the strength of the clinical impact of CD38 depletion, do the authors exclude a functional role of CD38 beyond maintaining the CNS Treg compartment?

We agree with this reviewer that the effects of losing antigen-specific Treg cells under conditions of CD38 deficiency are only partial at this stage of recovery. However, we would like to put forward two arguments that support the essential and cell-autonomous role of CD38 as a key determinant of stress resilience in

CNS Treg cells: first, the dynamics of MOG-tetramer binding T cells are opposite in conventional T cells and Treg cells when both lack CD38. Second, when CD38-sufficient and CD38-deficient Treg cells are present in the same environment, the demise of the CD38-deficient Treg cells is not rescued.

Based on our data, we cannot exclude additional roles of CD38 during CNS inflammation. For instance, CD38 has been shown to play a role in astrocytes, influencing neuroinflammation (Meyer et al., 2022). Here, CD38 appears to regulate intracellular (not extracellular) NAD⁺.

If CD38 works via preserving CD25 signalling, would enhancing the local IL2 levels in the CNS rescue the loss of Tregs observed in a CD38 KO?

This is a very interesting speculation. It is possible that increasing the amounts of IL-2 available in the CNS during recovery, when the conventional T cell population has contracted and no longer provides substantial amounts of this cytokine, would rescue the lack of CD38. Approaches involving synthetic biology or pharmacological administration of IL-2 in vivo have shown the potential to expand Treg cells in vivo (Yshii et al., 2022). However, this was mostly done in steady state and not under inflammatory conditions. The ADP-ribosylation of CD25 is relatively long-lived, and it must be a concern that the ADP-ribosylated IL-2 receptor may not respond appropriately, even if high amounts of IL-2 are provided exogenously.

Reviewer #3:

Major Concerns

1. The clinical score curves used after DTx treatment (such as in Figure 2d) are unusual for the field. It would be helpful if the authors could show one example full EAE score curve from Day 0 to last time point, to visualize how the DTx i.c. treatment impacts clinical scores. Additionally, histology images from DTx i.c. treated vs control mice would help validate that there is increased inflammation after local Treg depletion. This could be included in supplemental data.

In the revised version of the manuscript, we have included the full EAE scores of Fig. 2d, which show a matched disease course of those mice that were later depleted of local CNS Treg cells or were control-treated. We present these data together with an identical data set, in which the follow-up after local Treg-depletion was 10 days instead of 3 days in the new Supplementary Fig. 3.

According to this reviewer's suggestion, we performed histopathologic analyses on the CNS tissue of EAE mice in EAE recovery. While stable immune niches were observed after control treatment or systemic Treg cell depletion, local CNS-specific Treg cell depletion triggered an extensive inflammatory reaction in the CNS that appeared to originate from those immune niches and then extended into the parenchyma. These novel data are now included in the new Fig. 3 of the revised manuscript.

Minor Concerns

1. The authors use the term "MOG-activated" Tregs in the Fig. 4b experimental setup. They might wish to state that the experimental setup in Fig 4b "enriches" for MOG reactive Tregs, as the vast majority of transferred Tregs from the spleen in this setup will not be MOGreactive.

We concur with this point of view and amended the text of the manuscript accordingly (p. 13 of the revised manuscript).

2. On page 18, the authors argue that the effect of CD38 on Tregs was more pronounced in the antigen-specific cells based on their tetramer data. This is likely the case. In addition, tetramer staining intensity correlates with TCR affinity for a pMHC complex, and it has been shown that many non-MOG tetramer binding T cells in the CNS of EAE are have TCRs with low affinity for MOG(35-55)I-Ab. Thus, the authors may wish to mention that CD38 could also be more important on Tregs with a TCR that is high-affinity for MOG(35-55)I-Ab.

We agree with this statement and included a comment along those lines in the revised version of the manuscript (p. 15 of the revised manuscript)

3. The authors note and provide compelling data for negligible exchange of “inflammationtolerant CNS Tregs” with the periphery. To further support these data, they may wish to cite Wakim et al., PNAS (2010) which similarly showed that CD8+ T resident memory cells in the CNS adapt to the CNS environment, remain for long periods even without cognate antigen, and rapidly die when removed from the CNS milieu.

The biology of tissue resident memory T cells and tissue resident Treg cells might indeed be similar in many ways, and we are happy to cite Wakim et al., who were the first to identify Trm cells in the CNS (see p. 12 of the revised manuscript). In terms of Treg cells, however, ongoing cognate stimulation of CNS Treg cells, which reside in the tissue for extended periods of time, might indeed be very important, as indicated by the strong enrichment trajectory for MOG/IAb-tetramer binding Treg cells in the CNS over time.

References

- Hiltensperger, M., E. Beltran, R. Kant, S. Tyystjarvi, G. Lepennetier, H. Dominguez Moreno, I.J. Bauer, S. Grassmann, S. Jarosch, K. Schober, V.R. Buchholz, S. Kenet, C. Gasperi, R. Ollinger, R. Rad, A. Muschaweckh, C. Sie, L. Aly, B. Knier, G. Garg, A.M. Afzali, L.A. Gerdes, T. Kumpfel, S. Franzenburg, N. Kawakami, B. Hemmer, D.H. Busch, T. Misgeld, K. Dornmair, and T. Korn. 2021. Skin and gut imprinted helper T cell subsets exhibit distinct functional phenotypes in central nervous system autoimmunity. *Nat Immunol* 22:880–892.
- Meyer, T., D. Shimon, S. Youssef, G. Yankovitz, A. Tessler, T. Chernobylsky, A. Gaoni-Yogev, R. Perelroizen, N. Budick-Harmelin, L. Steinman, and L. Mayo. 2022. NAD(+) metabolism drives astrocyte proinflammatory reprogramming in central nervous system autoimmunity. *Proc Natl Acad Sci U S A* 119:e2211310119.
- Yshii, L., E. Pasciuto, P. Bielefeld, L. Mascali, P. Lemaitre, M. Marino, J. Dooley, L. Kouser, S. Verschoren, V. Lagou, H. Kemps, P. Gervois, A. de Boer, O.T. Burton, J. Wahis, J. Verhaert, S.H.K. Tareen, C.P. Roca, K. Singh, C.E. Whyte, A. Kerstens, Z. Callaerts-Vegh, S. Poovathingal, T. Prezzemolo, K. Wierda, A. Dashwood, J. Xie, E. Van Wonterghem, E. Creemers, M. Aloulou, W. Gsell, O. Abiega, S. Munck, R.E. Vandenbroucke, A. Bronckaers, R. Lemmens, B. De Strooper, L. Van Den Bosch, U. Himmelreich, C.P. Fitzsimons, M.G. Holt, and A. Liston. 2022. Astrocyte-targeted gene delivery of interleukin 2 specifically increases brain-resident regulatory T cell numbers and protects against pathological neuroinflammation. *Nat Immunol* 23:878–891.